# Cytoplasmic Inter-Subunit Interface Controls Use-Dependence of Thermal Activation of TRPV3 Channel

**DOI:** 10.3390/ijms20163990

**Published:** 2019-08-16

**Authors:** Lucie Macikova, Lenka Vyklicka, Ivan Barvik, Alexander I. Sobolevsky, Viktorie Vlachova

**Affiliations:** 1Department of Cellular Neurophysiology, Institute of Physiology Czech Academy of Sciences, 142 20 Prague, Czech Republic; 2Department of Physiology, Faculty of Science, Charles University, 128 00 Prague, Czech Republic; 3Division of Biomolecular Physics, Institute of Physics, Faculty of Mathematics and Physics, Charles University, 121 16 Prague, Czech Republic; 4Department of Biochemistry and Molecular Biophysics, Columbia University, New York, NY 10032, USA

**Keywords:** transient receptor potential, transient receptor potential vanilloid 1 (TRPV1), noxious heat, ankyrin repeat, nociception

## Abstract

The vanilloid transient receptor potential channel TRPV3 is a putative molecular thermosensor widely considered to be involved in cutaneous sensation, skin homeostasis, nociception, and pruritus. Repeated stimulation of TRPV3 by high temperatures above 50 °C progressively increases its responses and shifts the activation threshold to physiological temperatures. This use-dependence does not occur in the related heat-sensitive TRPV1 channel in which responses decrease, and the activation threshold is retained above 40 °C during activations. By combining structure-based mutagenesis, electrophysiology, and molecular modeling, we showed that chimeric replacement of the residues from the TRPV3 cytoplasmic inter-subunit interface (N251–E257) with the homologous residues of TRPV1 resulted in channels that, similarly to TRPV1, exhibited a lowered thermal threshold, were sensitized, and failed to close completely after intense stimulation. Crosslinking of this interface by the engineered disulfide bridge between substituted cysteines F259C and V385C (or, to a lesser extent, Y382C) locked the channel in an open state. On the other hand, mutation of a single residue within this region (E736) resulted in heat resistant channels. We propose that alterations in the cytoplasmic inter-subunit interface produce shifts in the channel gating equilibrium and that this domain is critical for the use-dependence of the heat sensitivity of TRPV3.

## 1. Introduction

Transient receptor potential vanilloid 3 (TRPV3) is an ion channel implicated in the regulation of skin homeostasis, thermo-sensing, and nociception [1,2,3,4,5]. Its temperature sensitivity is strongly use-dependent which means that the activation threshold and the steepness of temperature responses progressively and irreversibly decrease as the channel is repeatedly activated [4,6,7,8]. This enables TRPV3 to change from a transducer of extremely noxious heat (>50 °C) to a non-noxious mild warmth sensing detector (<33 °C). Previous studies have identified several sites in temperature-sensitive vanilloid TRP channels from which the thermal threshold and the steepness of temperature dependence can be affected [9,10,11,12,13,14,15,16,17]. Two transferable mutually exclusive regions have been proposed, each as a bona fide protein domain conferring the characteristic steep temperature dependence. The first is the pore domain identified in TRPV1 that is capable of conferring specific heat sensitivity when transplanted into the Shaker Kv channel [16]. The second region responsible for the strong temperature dependence of all temperature-sensitive vanilloid family homologues is the membrane proximal domain (MPD) connecting the N-terminal ankyrin repeat domain (ARD) to the first transmembrane segment S1 [18]. An insertion of a single specific amino acid residue into the MPD linker region (V408–L422) results in a less temperature stable conformation of TRPV3 [8]. While these previous studies have seemingly narrowed down the discrete domains mediating temperature sensitivity, heat responses can also be dramatically affected by as little as a single residue mutation in the N-terminal ARD remotely located from the two aforementioned regions [17]. Clearly, there is still no definitive answer as to whether thermally activated channels use a local, domain-based, mechanism for sensing the temperature, or whether a global response, distributed over the channel structure, is underlying their strong thermal sensitivity [19,20,21,22]. 

The crystal structure of the cytoplasmic N-terminal ARD of TRPV3 isolated from the rest of the protein exhibits a stable conformation of a loop between ankyrin repeats 3 and 4, called finger 3 [23] (Figure 1A–C). Whereas the finger 3 is a flexible region in a majority of vanilloid receptor ARD structures, in TRPV3-ARD (Protein Data Bank Code, PDB ID, 4N5Q) it bends over the inner helices of the ankyrin repeats 3 and 4 and is stabilized by hydrogen bonds and hydrophobic packing [23], indicating its unique structural role in TRPV3. An increasing number of structures of the vanilloid TRP channels show that the region around the finger 3 forms a key interface between the ARD and MPD of the adjacent subunit (Figure 1C). The newest cryo-electron microscopic structures of the full length TRPV3 captured in various states along the activation cycle reveal pronounced inter-subunit interfaces formed by the 3-stranded β sheet and the C-terminal loop of one subunit and the ARD repeats 2–5, including finger 3, of the adjacent subunit [24,25]. Since the interface-forming domains are critically involved in temperature sensitivity of TRPV3 and its close relatives [10,17,18,26,27], we reasoned that the unique properties of finger 3 might be important for setting the thermal threshold for TRPV3 activation. We were curious whether substitutions in finger 3 with the homologous residues of TRPV1 can change the thermal threshold of TRPV3 to resemble the one of TRPV1.

## 2. Results

### 2.1. Heat Activation of Wild-Type Human TRPV3 Is Strongly Use-Dependent

Previous studies have shown that the initial activation of mouse TRPV3, under certain experimental conditions, requires temperatures above 50 °C and the activation threshold and the slope sensitivity of the temperature dependence decrease upon repeated stimulation [8]. To confirm this characteristic activation pattern for human TRPV3, we first measured heat responses in control bath solution using three repetitive temperature ramps from 25 °C to either below or above 50 °C, applied at a maximum speed of about 35 °C/s at 3 s intervals (Figure 2A–F). To assess the maximum activation capacity of the channels, the cells were stimulated subsequently three times by heat in the presence of a combination of the two TRPV3 activators, 2-APB and carvacrol (100 µM and 100 µM). In extracellular control bath solution, repeated stimulation to ~48 °C produced only very small currents (154 ± 21 pA; *n* = 9) with weak temperature dependence over the range of 45–48 °C (*Q*_10_, temperature coefficient of 2.8 ± 0.3) (Figure 2A,C). In contrast, the agonists produced robust currents at room temperature (5.3 ± 0.3 nA at 25 °C; *n* = 9) that became saturated at ~40 °C (9.8 ± 1.7 nA) and after washout and cessation of the heat stimulus decayed to the basal level. Subsequent reapplication of heat in control bath solution produced again only small responses. When the maximum value of the initial temperature stimulus was raised above ~50 °C, the activation pattern was quite different (Figure 2B,D). Untransfected HEK293T cells did not exhibit any sensitivity to heat (>60 °C; Appendix A) or the combination of agonists (Appendix A). The initial TRPV3-mediated heat responses reached peak amplitudes of 1.7 ± 0.6 nA (*n* = 25) and, upon repeated exposure to heat, the threshold for activation decreased while the maximum amplitude of currents increased (Figure 2E), indicating that high temperatures above 50 °C activated the channels specifically and robustly. To describe the effects of temperature on membrane currents, we evaluated the maximum apparent temperature coefficients *Q*_10_ and the thermal thresholds from the slope of Arrhenius plot (absolute values of inward currents plotted on a logarithmic scale, *y*-axis, against the reciprocal of the absolute temperature, *x*-axis), as described previously [27,29]. It should be noted that the gating kinetics of TRPV3 is unknown and involves irreversibility [8]. The apparent *Q*_10_ and threshold values thus reflect complex effects of temperature on channel gating kinetics and the applicability of such characteristics is discussed in detail by Liu and Qin [8,26]. By fitting the Arrhenius plot of the current-temperature relationships, the initial heat-induced responses exhibited the median *Q*_10_ value of 22.2 (first and third quartile of 17.1 and 33.9) over the temperature range from 51.7 ± 1.2 °C to 53.9 ± 1.1 °C. The median *Q*_10_ value of the subsequent heat response was lowered significantly to 10.6 (first and third quartile of 7.2 and 12.6; *n* = 14; *p* = 0.004, paired *t*-test) without changes in the temperature range for activation (from 52.5 ± 1.1 to 55.7 ± 1.2 °C). Over the low temperature range (25.9 ± 0.4 to 34.4 ± 1.2 °C), the average *Q*_10_ of initial heat responses was 1.8 ± 0.1 and this value was not different from *Q*_10_ of the second response (1.6 ± 0.1; *p* = 0.222). 

From the current–temperature relationships of the initial heat responses it was apparent that the estimated threshold for heat activation and the *Q*_10_ values varied substantially from cell to cell (Figure 2F). We attributed this to unavoidable variations in the maximum speed of temperature change because these values showed strong correlation across the cells (Appendix A). Previously, Qin’s group [8] demonstrated that the rate of transition between the closed and open states of mouse TRPV3 is primarily driven by temperature and that the direct measurement of the activation enthalpy requires a special heat stimulator enabling to exchange temperature at a very high speed (55 °C/ms). Most likely, the slower rate of temperature change in our case (35 ± 1 °C/s; *n* = 22; see Materials and Methods) allowed the activated channels, to a variable extent, to transit to the state with a reduced energetics of gating. Despite this considerable variability, the currents that pooled together from the individual cells and normalized to their respective maximum amplitudes were clearly and statistically distinguishable from the heat-induced currents through rat TRPV1 (Appendix A). This normalization procedure (see Materials and Methods) consisted of pooling the data from all the cells across the *x*-axis (temperatures 26–64 °C) and *y*-axis (0–1) and was used only for synoptic comparisons because a simple pooling of individual Arrhenius plots or normalizing the initial currents at room temperature (as e.g., previously used for TRPV1 by Zhang et al. [16]) did not capture the effects on all of the complex temperature-dependent and use-dependent channel behavior. Together, these data corroborate previous findings [8] that TRPV3 is activated at high noxious temperatures, its initial activation threshold exceeds 50 °C, and the channel is strongly use-dependent. 

### 2.2. Mutations in the Tip of Finger 3 Alter the Threshold and Steepness of TRPV3 Temperature Dependence

Using a chimeric approach, we replaced the residues within the tip of the finger 3 of human TRPV3, either individually or in combinations, by cognate residues from rat TRPV1. We generated one septuplet mutant TRPV3/V1(251–257) in which the sequence NPKYQHE was replaced with KKTKGRP and two chimeric triplet mutants in which the residues over the regions 251–254 and 255–257 were replaced by those of TRPV1 (see Figure 1A). We examined the temperature sensitivity of all the constructs by comparing their initial heat responses. Example traces and the data from these experiments are summarized in Figure 3A–K and Appendix A. 

Consistently, heat responses through the septuplet mutant were initially sensitized and exhibited tonic activity at room temperature (Figure 3A). The plot of the normalized current–temperature relationships of the heat responses exhibited a steeper slope at temperatures around 40 °C and a shallower slope at higher temperatures (Figure 3C and Appendix A). Also, the TRPV3/V1(251–254) triple mutant exhibited a clearly sensitized phenotype (Figure 3B,D) so that the maximum amplitude of the initial heat responses measured in control extracellular solution (10.1 ± 1.3 nA) were not different from those recorded in the presence of the combination of agonists (11.2 ± 1.1 nA; *p* = 0.200, paired *t*-test; *n* = 13). This was in apparent contrast with the maximum heat responses of wild-type channels (Figure 2B) that increased from 1.9 ± 0.5 to 10.3 ± 0.9 nA in the presence of agonists (*p* ≤ 0.001, paired *t*-test; *n* = 19). In contrast to wild-type channels, the *Q*_10_ values of the initial heat responses were not significantly different from *Q*_10_s of the subsequent heat responses (median values of 14.5 and 15.5; *p* = 0.804, paired *t*-test; *n* = 11). The currents through the triple mutant TRPV3/V1(255–257) were not different from wild-type channels (Figure 3E). The individual substitutions of residues N251, P252, K253, and Y254 further indicated that the strong impact of the septuplet mutation on the activation threshold was primarily due to changes in the size or flexibility of proline at position 252 and, to a lesser extent, lysine at position 253 (Figure 3F–K and Appendix A). To compare the sensitization propensity and the threshold for specific activation among all the mutants, we evaluated the maximum *Q*_10_ and the temperature range over which the Arrhenius plot was nearly perfectly linear (correlation coefficient *r*^2^ of 0.98–0.99) (see Material and Methods and Figure 4A,B). One-way ANOVA clearly confirmed a highly significant (*p* < 0.001) decrease in maximum *Q*_10_ values and activation thresholds of currents through the septuplet mutant and the chimeric triple mutant over the positions 251–254, but not of the currents through the triple mutant over the positions 255–257 (Figure 4C–E). To further perturb the finger 3 loop, we introduced pairs of cysteine residues at positions that are structurally predicted to lie in close proximity. The aim was not only to disrupt the existing network of interactions shaping the finger loop but also to examine whether these cysteines can form a disulfide bridge during heat stimulation and thus affect the channel responsiveness. The double mutation P252C/E257C significantly reduced the maximum steepness of the temperature dependence (*Q*_10_; *p* = 0.004; *n* = 5), whereas K246C/E263C lowered the temperature range of the steepest slope (*p* = 0.04, *n* = 4; Appendix A). Currents produced by single cysteine mutations P252C and E257C were not statistically different from wild-type TRPV3, indicating that intra-subunit interactions within the finger 3 loop are only partly responsible for the sensitized phenotype of TRPV3. Because the P252G and P252C mutants exhibited the wild-type phenotype (Figure 3H and Appendix A), the size and/or flexibility of the residue at this position seem to be crucial for maintaining normal functioning of the channel. Together, these data point to the tip of the finger 3 as a region with a great deal of influence over the TRPV3 temperature dependence characteristics, including the threshold for activation and the steepness of heat responses. 

### 2.3. Substituted Residues in Finger 3 Do not Directly Interact with the MPD

Superposition of the open state structures of TRPV3 (PDB ID: 6DVZ) and TRPV6 (6D7S) suggests that the stretch of lysine residues introduced into the tip of the finger 3 in TRPV3 can potentially contact the N412-D414 loop region of the MPD via salt bridge interactions analogous to those seen between R154, E250, and K301 in TRPV6 [30]. Because the MPD loop region strongly influences the use dependence of TRPV3 [8], we next explored whether the sensitizing effects of mutations in finger 3 can result from possible interactions with D414. Neutralization of D414 by alanine did not affect the heat-induced responses (Appendix A). This result is in consonance with the previous findings of Liu et al. [8], who did not observe any functional changes when D414 was substituted with proline. Channels with combined mutations P252K/D414A, N251K/D414A, and TRPV3/V1(251–254)/D414A phenocopied characteristics of the parent templates (Appendix A), arguing against the direct interactions between the tip of the finger 3 and the MPD loop to be responsible for the sensitized phenotypes of the chimeric mutants.

### 2.4. Inter-Subunit Interface Controls TRPV3 Gating

Several recent structures of full-length TRPV channels [24,25,31,32,33,34,35] reveal that finger 3 contributes to the inter-subunit interface between the ARD and the three-stranded β-sheet composed of a β-hairpin from the N-terminal MPD linker and a C-terminal β-strand, both from the adjacent subunit. In TRPV3, ankyrin repeats 2–5 of the ARD are additionally contacted by the C-terminal loop domain that follows the C-terminal β-strand and wraps around the β-sheet [24,25]. Structural comparisons of the recently published structures of TRPV3 [24] in the closed and open states indicate that the β sheet-ARD finger 3 interface between neighboring subunits changes during gating—the contacting regions „slide“ relative to each other upon channel opening/closure (Figure 5). To explore the role of the inter-subunit interface in temperature dependent activation, we further measured heat-induced responses from double cysteine mutants designed to crosslink the interface (Figure 6 and Figure 7, and Appendix A). The selected cysteine pairs showed substantial differences in distances between their Cα atoms in the closed and open state structures: F259C/Y382C (8.3 Å in the closed state and 9.6 Å in the open state), F259C/V385C (6.6 Å and 7.3 Å), and H256C/E736C (7.3 Å and 8.1 Å). We hypothesized that using the disulfide-locking strategy we can capture the channels in the resting state and raise the activation energy of temperature-dependent gating. 

Indeed, compared to wild-type channels, the double cysteine mutant H256C/E736C showed very small current responses to temperatures near ~60 °C (0.4 ± 0.2 nA; *p* = 0.01; *n* = 8) (Figure 6A–C). We were able to dissect and quantify the parameters of specific heat activation in only 3 out of 8 cells (Figure 7C), whereas the remaining cells exhibited small nonspecific currents over the entire range of examined temperatures (Appendix A). Significantly smaller instantaneous currents compared to wild type were also mediated by the H256C/E736C mutant at room temperature in response to the combination of agonists (1.8 ± 0.6 nA; *p* ≤ 0.001; *n* = 8). Upon heating, however, these responses increased dramatically and their amplitudes were significantly larger than even the maximum responses of wild-type channels (16.1 ± 1.0 nA; *p* = 0.002; Appendix A). This result suggests that the double cysteine mutant H256C/E736C is sensitive to heat but its apparent threshold for heat activation is shifted beyond the testable temperature range unless the mutant is sensitized with agonists. In a surprising contrast, the cells expressing F259C/Y382C and F259C/V385C exhibited large leakage-like inward currents (Figure 6D,E, and Appendix A) with almost linear current–temperature relationships over the entire experimental temperature range (Figure 6F). Particularly, cells expressing F259C/V385C were so deleteriously leaky at the holding potential of –70 mV (4.4 ± 1.1 nA; *n* = 5) that we were unable to complete our recordings in ~75% of the cells examined. To rule out the possibility that these large currents represented nonspecific leakage, we applied the TRPV3 inhibitor ruthenium red (10 µM) at the end of each recording. As shown in Figure 6D,E, and Appendix A, ruthenium red blocked the currents completely, indicating that the double cysteine mutant channels were locked in the open conformation. There were no significant changes in the maximum heat responses recorded from F259C/V385C and F259C/Y382C in the presence of a combination of agonists compared to wild-type channels (Appendix A; *p* = 0.724 and 0.318; *n* = 4 and 6), suggesting that the double cysteine mutations interfere with channel gating rather than the channel expression level. The individual cysteine substitutions F259C, Y382C, and V385C only mildly influenced the temperature dependence properties of the channels (Figure 7C and Appendix A). 

To further verify that the observed effects are indeed caused by disulfide bond formation, we pretreated cells expressing the F259C/V385C construct with the membrane permeable reducing agent dithiothreitol (DTT; 5 mM) for 15–20 min at 37 °C. DTT was then washed off with bath solution before experiments. In clear contrast to the observations made for untreated cells, the currents through the F259C/V385C channels were not tonically activated after whole-cell formation and exhibited specific sensitivity to heat (Figure 6G,H; Figure 7C,E; and Appendix A). The initial heat responses reached peak amplitudes of 5.8 ± 1.5 nA and exhibited the apparent *Q*_10_ values somewhat lower than wild-type channels (9.0 ± 1.2; *p* = 0.012, *n* = 5) over the temperature range from 46.9 ± 1.2 to 52.6 ± 1.7 °C. We did not observe any significant changes in the heat-induced responses in control untreated cells expressing wild-type channels. This result confirms the proximity of F259 and V385 and suggests a specific effect of the inter-subunit interface crosslinking on the channel gating equilibrium.

In contrast to the prominent effects seen for F259C/V385C, the 15–20-min pretreatment with DTT had no impact on the H256C/E736C channels. The overall activation pattern apparently resembled the untreated H256C/E736C expressing cells in that only very small nonspecific currents were induced by temperatures up to ~60 °C (0.17 ± 0.02 nA; *n* = 6) (Appendix A). The currents measured in the presence of agonists were small at room temperature and were strongly sensitized by heat above ~50 °C (to 14.0 ± 1.7 nA). This result may suggest that the H256C/E736C disulfide bond is so much favorable and protected (e.g., sterically) that the amount of DTT that can reach it is insufficient to break it. Alternatively, the observed changes are due to individual substitutions of H256 or E736. To resolve this issue, we next measured responses from E736C in the control bath solution and observed the same heat-resistant phenotype as seen for H256C/E736C (Appendix A): currents measured in the presence of agonists were small at room temperature (0.6 ± 0.2 nA) but were strongly sensitized by heat above ~50 °C (to 16.7 ± 1.1 nA; *n* = 4) (Appendix A). Given that the substitution of H256 with arginine within the triplet TRPV3/V1(255–257) did not affect channel function (Figure 3E), the observed effects of the double cysteine substitution H256C/E736C are most likely the result of the mutation of E736. 

### 2.5. Molecular Dynamics Simulations Reveal Temperature Sensitive Regions

To evaluate impact of heat on the inter-subunit interface, we subjected the mouse TRPV3 apo (closed) state structure (PDB ID: 6DVW) to molecular dynamics (MD) simulations at three temperatures, 300, 330, and 400 K (Figure 8A,B and Appendix A). The root mean square deviation (RMSD) along the 50-ns MD trajectory indicated that the models were stable (Appendix A). At 300 K, the per-residue RMSD calculated for the backbone atoms identified several highly flexible regions (RMSD > 4 Å), including the finger 3 loop, S2-S3 linker, and the C-terminal region C721-D727 (Figure 8A). Interestingly, the latter region did not exhibit high flexibility at 400 K (Appendix A). The ARD finger 3 and finger 4 loops that are flexible at 300 K became even more flexible at 400 K. Therefore, apart from the S2–S3 linker, the C-terminal loop domain, the finger 4 loop, and the finger 3 loop in particular appear to be regions that are most flexible and most sensitive to heat. These are the regions that in the context of the full-length channel either directly form (finger 3 loop) or are immediately adjacent to (C-terminal loop domain and finger 4 loop) the β sheet-finger 3 inter-subunit contact. The results of MD simulation experiments therefore strongly implicate the β sheet–finger 3 inter-subunit interface into TRPV3 regulation by temperature and support our mutagenesis experiments. Interestingly, the β-sheet, MPD and TRP helix, which link the β sheet–finger 3 inter-subunit interface to the channel gate at the S6 bundle crossing, appear to be the least flexible in our MD simulations and more so at high temperatures. We therefore hypothesize that these three domains form a rigid gating transmission element that communicates changes at the inter-subunit interface directly to the channel gate. Such changes, which might be caused by the application of heat/cold or by introducing cross interface disulfide links, are capable of shifting the conformational gating equilibrium between the open and closed states.

## 3. Discussion

Our results indicate that interaction of the ARD finger 3 tip (region around H256-F259) with the neighboring subunit β-sheet (via Y382 and to a larger extent V385 in the β1–β2 hairpin) can be altered to lock the TRPV3 channel in its open state and abolish its steep temperature dependence. On the other hand, mutation at the conserved E736 in the β3 strand of the β-sheet shifts the threshold for heat activation to higher temperatures. We propose that the direct interaction of the tip of the finger 3 with the 3-stranded β-sheet of the adjacent subunit is responsible for the TRPV3 sensitized phenotype. This hypothesis is supported by our analysis of basal inward currents measured from all the mutants at −70 mV which distinctly reveals tonically active phenotypes (Appendix A). Apart from the cross-interface disulfide links, the strongest energetic effects on the basal closed-to-open equilibrium were observed for V3/V1(251-254), P252K, K253T, and combinations of these mutations with D414A. The region of the MPD harboring D414 was previously implicated in use dependence of TRPV3 [8]. Since D414A mutation alone exhibited wild-type phenotype, the signal from finger 3 might need a proper transmission through this critical portion of the MPD. Our structural comparisons (Figure 5) indicate that the key β-sheet-ARD finger 3 inter-subunit interface undergoes significant changes between the closed and agonist-activated open states [24] due to sliding of the contact regions relative to each other. Correspondingly, our data suggest that alterations in this interface do in turn produce shifts in the channel gating equilibrium. 

How conformational changes within the tip of the finger 3 and in the key inter-subunit interface are transmitted to the gate and how the sensitized state is imposed on the channel? Our MD simulation results show that the ARD finger 3 is highly dynamic and more so at higher temperatures (Figure 8A and Appendix A). This also makes the β-sheet–ARD finger 3 interface highly dynamic, which strengthens the conclusion of the Figure 5 structural comparison. More importantly, our MD simulation results suggest that the entire region between this interface and the gate is rigid (β-sheet, MPD and TRP helix, see Figure 8B). We propose that due to its rigidity, this gating transmission element can quickly and efficiently translate heat-induced changes at the inter-subunit interface to the opening or closure of the ion channel. Further MD simulations will be necessary to test this hypothesis. 

Are there other structural elements that can contribute to TRPV3 regulation by temperature? Recent extensive MD simulations performed with the TRPV3-related TRPV1 channel revealed various key domains that undergo conformational changes in response to the increase in temperature [36,37,38,39,40,41]. These studies support several lines of experimental evidence that the channel can be opened by temperature after a series of conformational changes that propagate from the peripheral regions to the channel pore. This process is contributed by the S1–S2, S2–S3 and S4–S5 linkers, the MPD, the distal C-terminus, the TRP helix, and the S5–S6 pore domain. The largest heat-activated motions were observed in the S1–S2 and S2–S3 linkers and thus conformational changes in these domains have been hypothesized to represent early structural events that prime the channel for opening [39]. These observations are not surprising in view of the recently published structures of TRPV3 in the closed apo and agonist-bound open states [24]. The structures indicate that conformation of the S1–S2 linker is coupled to the occupancy of the lipid binding site #1 which might strongly depend on temperature. In addition, the S2–S3 linker makes contacts with the TRP helix and the MPD and through these interactions can affect conformation and relative position of the TRP helix, which is directly connected to the gate (Figure 5). This concept does not contradict another possible mechanism of thermal actuation in which heat induces ejection of phosphatidylinositol lipids from the vanilloid binding pocket formed by S3, S4 and the S4–S5 linker of one subunit, and S5 and S6 of the adjacent subunit [41,42]. Stability of the conductive conformation may further depend on hydration of the channel pore [43].

Recent unbiased and high-throughput scanning mutagenesis studies of TRPV1 [9] lent support to the previously proposed heat capacity mechanism in which the collective importance of hydrophobicity and the distributed nature of temperature sensitive structures have been predicted [20,22]. In our experiments, strong effects of mutations on TRPV3 temperature regulation were not accompanied by significant changes in hydrophobicity. While we cannot rule out contribution of the heat capacity mechanism to temperature regulation of thermosensitive TRP channels, our results indicate that the ARD-β sheet inter-subunit interface is a critical element of the temperature-sensing molecular machinery of TRPV3.

During the preparation of this manuscript, an article was published [44], reporting that TRPV3 gating induced by chemical agonist 2-APB involves large rearrangements at the cytoplasmic inter-protomer interface and that this motion triggers coupling between cytoplasmic and transmembrane domains, priming the channel for opening. Thus, our data strongly indicate that the inter-subunit interface is critical not only for agonist-induced but also for heat-induced gating of TRPV3.

## 4. Materials and Methods 

### 4.1. Cell Culture, Mutagenesis and Transfection of HEK293T Cells

HEK293T cells were cultured in Opti-MEM I medium (Invitrogen) supplemented with 5% FBS. The day before transfection, cells were plated in 24-well plates (2 × 10^5^ cells per well) in 0.5 mL of medium and became confluent on the day of transfection. The cells were transiently co-transfected with 300 ng of plasmid encoding wild-type or mutant human TRPV3 (in pcDNA5/FRT vector, kind gift of Prof. Ardem Patapoutian, The Scripps Research Institute, San Diego, USA), and with 200 ng of GFP plasmid (Takara, Shiga, Japan) using the magnet-assisted transfection technique (IBA GmbH, Goettingen, Germany) and then plated on poly-l-lysine-coated glass coverslips. Rat TRPV1 cDNA was kindly provided by David Julius (University of California, San Francisco, CA, USA). At least three independent transfections were used for each experimental group. The wild-type channel was regularly tested in the same batch as the mutants. The cells were used 24–48 h after transfection. The mutants were generated by PCR using a QuikChange II XL site-directed mutagenesis kit (Agilent Technologies, Santa Clara, CA, USA) and confirmed by DNA sequencing (GATC Biotech, Konstanz, Germany). 

### 4.2. Patch Clamp Recording and Heat Stimulation

Whole-cell membrane currents were recorded by employing an Axopatch 200B amplifier and pCLAMP 10 software (Molecular Devices, Sunnyvale, CA, USA). Patch electrodes were pulled from borosilicate glass capillary with a 1.5-mm outer diameter (Science Products GmbH, Hofheim, Germany). The tip of the pipette was heat-polished, and its resistance was 3–5 MΩ. Series resistance was compensated by 50–70% after compensation of fast and slow capacitance. Only one recording was performed on any one coverslip of cells to ensure that recordings were made from cells not previously exposed to heat or chemical stimuli. The extracellular solution before the whole-cell recording contained: 160 mM NaCl, 2.5 mM KCl, 1 mM CaCl_2_, 2 mM MgCl_2_, 10 mM HEPES, 10 mM glucose, 320 mosmol, adjusted to pH 7.4 with NaOH. The extracellular control bath solution for recording contained: 150 mM NaCl, 5 mM EGTA, 10 mM HEPES, 300 mOsm, adjusted to pH 7.4 with NaOH. The pipette solution contained 140 mM CsCl, 1 mM EGTA and 10 HEPES (263 mOsm, pH 7.4 adjusted with CsOH). A system for rapid superfusion of the cultured cells was used for thermal stimulation and drug application [45]. Briefly, experimental solutions were driven by gravity from seven different barrels, through automatically controlled valves, to a manifold that consisted of fused silica tubes connected to a common outlet glass capillary. The lower part of the capillary was wrapped with densely coiled copper wire that heated the solution to a chosen final temperature. Voltage commands for heat and agonist stimulation were generated from Digidata 1440 digitizer using pCLAMP 10 software (Molecular Devices, Sunnyvale, CA, USA). Ramp-shaped temperature increases from room temperature to >50 °C within 1500 ms were applied at 3 s intervals. The volume of solution in the experimental dish and the immersion of the application capillary were maintained at a constant level and an internal table generated after balancing an electrical circuit of the system was utilized, which resulted in a good reproducibility of the heat stimuli. The average speed of temperature changes did not significantly differ for all of the mutants throughout all experiments and did not correlate with the observed average *Q*_10_ value changes (Appendix A). All chemicals were purchased from Sigma-Aldrich (Merck).

### 4.3. Molecular Dynamics Simulations 

For MD simulations, the apo structure of TRPV3 (PDB code 6DVW) was parametrized using the AMBER_ILDN force field [46] and surrounded with TIP3P water molecules [47]. Prior to production MD simulations, all systems were energy minimized using the *pmemd* module of AMBER 14 [48]. Production MD simulation runs (lasting for 50 ns) were performed with the *pmemd.cuda*.*MPI* module of AMBER 14, which runs exclusively on GPUs at the equivalent speed of tens of standard processor cores [49]. The SPFP precision model was used [49]. Periodic boundary conditions (PBC) were applied. The particle mesh Ewald (PME) was used for calculation of electrostatic interactions [50]. 9 Å cutoff distance was applied for Lennard–Jones interactions. The temperature was maintained at 300, 330, or 400 K via Langevin dynamics with a friction factor of 5. Further, the Monte Carlo barostat (a new addition to Amber 14), that samples rigorously from the isobaric–isothermal ensemble and does not necessitate computing the virial, was used for the production phase. Covalent bonds involving hydrogen atoms were constrained using the SHAKE algorithm. For water molecules, a special “three-point” SETTLE algorithm was used [51]. The hydrogen mass repartitioning scheme (the mass of the bonded heavy atoms to hydrogen is repartitioned among hydrogen atoms, leaving the total mass of the system unchanged) allowed a time step set to 4 fs [52]. By default in AMBER 14, partitioning is only applied to the solute since SHAKE on water is handled analytically (via the SETTLE algorithm). Data were recorded every 1000 ps. Trajectories analyses were performed using the *cpptraj* module of AMBER 14 [53] and using the UCSF Chimera software package [54].

### 4.4. Data Analysis

Electrophysiological data were analyzed in pCLAMP10 (Molecular Devices). Curve fitting and statistical analyses were done in SigmaPlot 10 and SigmaStat 3.5 (Systat Software). The heat-evoked whole cell currents sampled during the rising phase of temperature ramp were pooled for every 0.25 °C. The maximum apparent temperature coefficients *Q*_10_ and the thermal thresholds were determined from the slope of Arrhenius plot (absolute values of inward currents plotted on a logarithmic scale, *y*-axis, against the reciprocal of the absolute temperature, *x*-axis) as described previously [27,29]. *Q*_10_’s were determined by using the formula: *Q*_10_ = exp (Δ*T E*_a_/(*R T*1 *T*2)), where *R* is the gas constant, Δ*T* = 10 Kelvin, *E*_a_ is an apparent activation energy estimated from the slope of Arrhenius plot between absolute temperatures *T*1 and *T*2. The lower and upper limits for *Q*_10_ estimation were defined as the temperatures at which the fit of the Arrhenius plot declined significantly from a straight line (*r*^2^ < 0.98). For synoptic comparisons between wild-type and mutant channels, initial heat-induced responses were subtracted by currents produced at 26 °C, normalized to their maximum value and pooled across *x*-axis (26–64 °C) and *y*-axis (0–1) for each experimental group from at least three independent transfections. Statistical significance was determined using Student’s *t*-test or one-way analysis of variance followed by Dunnett’s post hoc, as appropriate. 

## Figures and Tables

**Figure 1 ijms-20-03990-f001:**
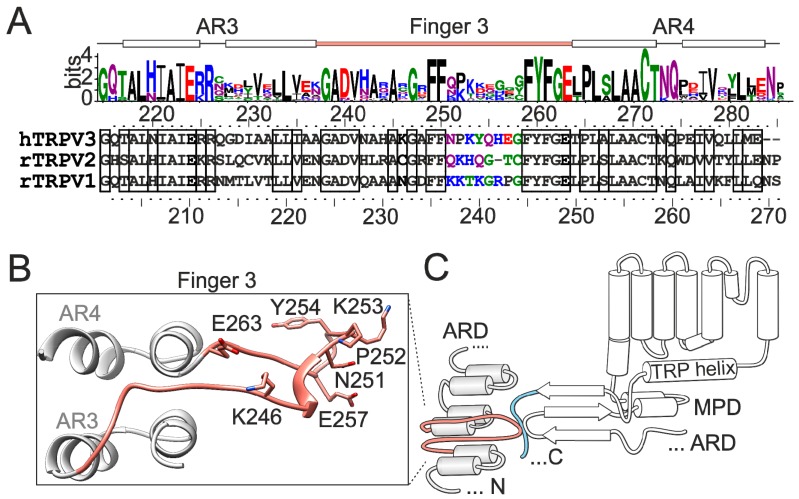
Schematic representation of the finger 3 loop region of transient receptor potential vanilloid 3-ankyrin repeat domain (TRPV3-ARD). (**A**) Sequence logo along thermally activated TRPV channels (fish, frog, snake, chicken, mouse, rat, and human TRPV1-4) and amino acid alignment of human TRPV3, rat TRPV2, and rat TRPV1 with elements of secondary structure indicated above. Positions are numbered according to human TRPV3 (above) and rat TRPV1 (below). The residues of the tip of the finger 3 are indicated in colors. (**B**) Residues within the finger 3 mutated in this study shown in the context of ankyrin repeats 3 and 4 (AR3 and AR4) of the mouse TRPV3 structure (Protein Data Bank Code 6DVW). (**C**) Topology diagram of TRPV3 N-terminal finger 3 (salmon) positioned in close proximity of the loop from the ankyrin repeat domain-transmembrane domain (ARD-TM) linker region (red) and the C-terminal loop domain (light blue) of an adjacent subunit. MPD, membrane proximal domain. The sequence logo was generated via WebLogo3 server [28].

**Figure 2 ijms-20-03990-f002:**
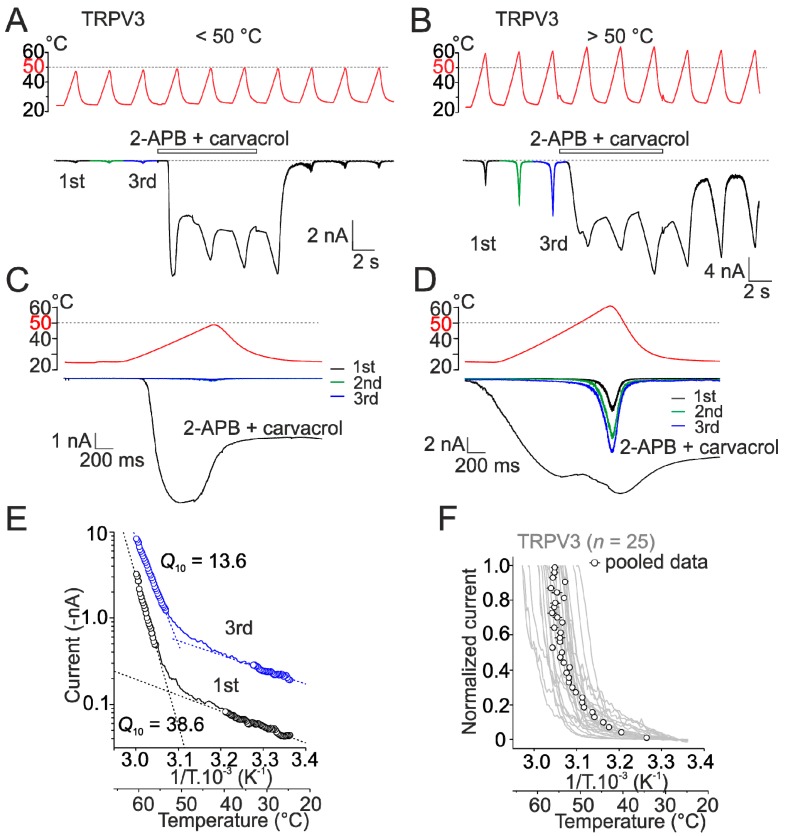
Activation of TRPV3 at temperatures above 50 °C. (**A**,**B**) Representative whole-cell currents evoked by heat stimuli below (**A**) or above (**B**) 50 °C, applied at −70 mV in wild-type TRPV3, in the absence or presence of mixed agonists (100 µM 2-APB + 100 µM carvacrol). (**C**,**D**) Expanded traces of the first to fourth heat responses (the fourth recorded in the presence of agonists) are shown below the recordings from A and B. (**E**) Comparison of the 1st and 3rd heat responses recorded in the absence of agonists from the cell shown in B. Arrhenius plot in which the current (*y*-axis, log scale) was plotted against the reciprocal of the absolute temperature (*x*-axis). The temperature coefficient, Q10, was estimated by linear regression (dashed lines) from the slope of the Arrhenius plot in the temperature ranges indicated by open circles. (**F**) Heat-induced currents measured from 25 cells expressing wild-type TRPV3, normalized to maximum responses. Pooled values of normalized heat-induced currents through wild-type TRPV3 (open circles with bi-directional grey bars indicating S.E.M.; *n* = 25). Currents from individual cells are overlaid for comparison (grey lines). In some cases, the error bars are smaller than the symbol.

**Figure 3 ijms-20-03990-f003:**
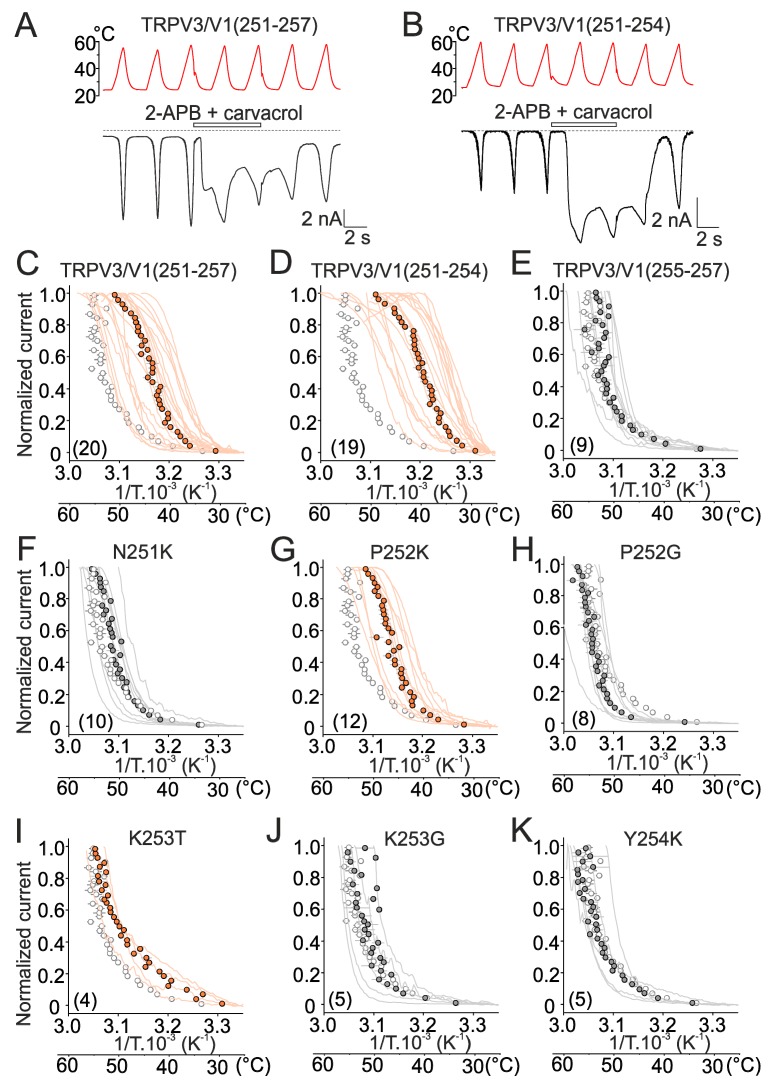
Mutations in the ARD finger 3 producing sensitized phenotypes. (**A**,**B**) Representative whole-cell recordings of currents evoked by repetitive heat stimuli from indicated chimeric TRPV3/V1 channels recorded in the absence or presence of mixed agonists (100 µM 2-APB + 100 µM carvacrol). Holding potential, −70 mV. (**C**–**K**) Heat responses from HEK293T cells expressing the indicated mutants of the finger 3, normalized to the maximum amplitude with average values overlaid (filled circles with gray error bars indicating S.E.M.). The average current for wild-type TRPV3 is overlaid for comparison as empty circles with grey bars indicating mean ± S.E.M. The number of cells is indicated in parentheses.

**Figure 4 ijms-20-03990-f004:**
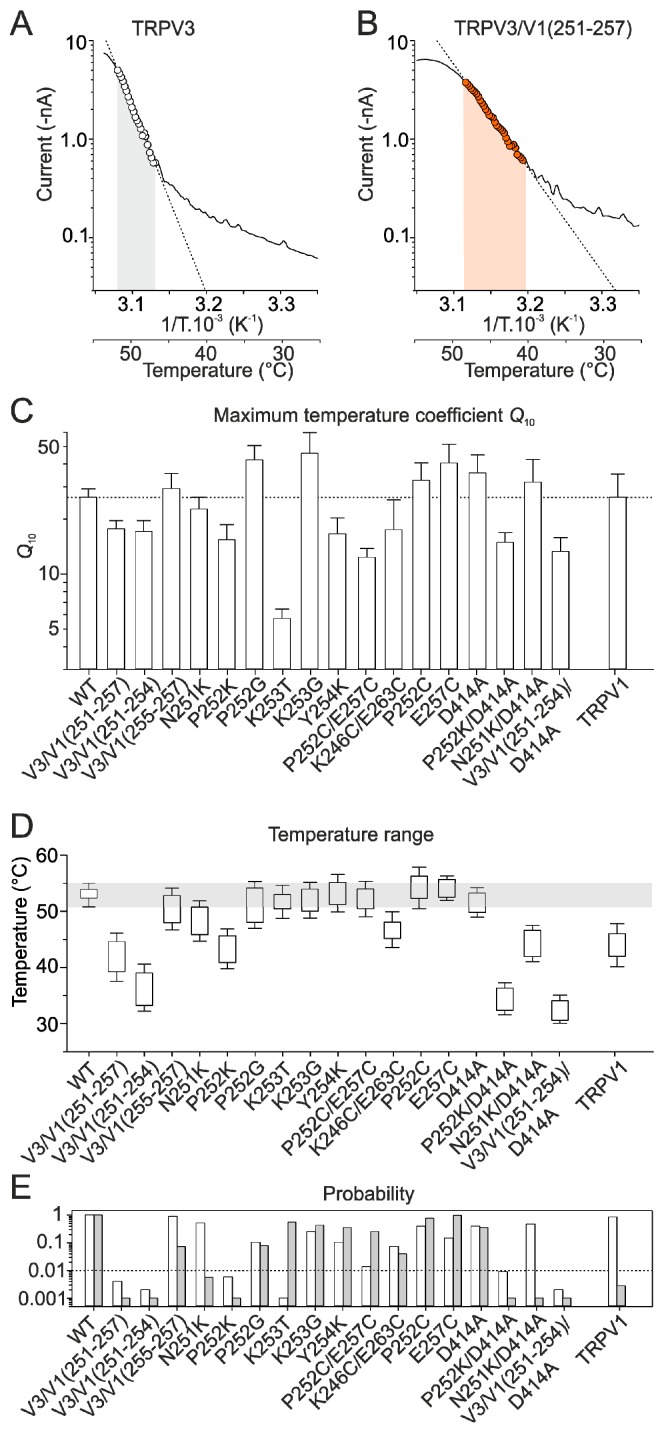
Statistical evaluation of maximum apparent *Q*_10_ and the estimated temperature thresholds for all the mutants. (**A**,**B**) Arrhenius plot of heat responses obtained from two representative cells expressing either wild-type (**A**) or mutant (**B**) channels. Temperature coefficient *Q*_10_ was determined for the initial heat response of each cell over the temperature range at which the Arrhenius plot was linear (dashed lines, *r*^2^ = 0.98–0.99). Temperature ranges used for the estimation of *Q*_10_ are indicated (shadow area). (**C**) Summary of the maximum temperature coefficient *Q*_10_ values for all the measured chimeras and mutants, compared with wild-type TRPV3 and rat TRPV1. The horizontal dotted line indicates average maximum *Q*_10_ of wild-type TRPV3. (**D**) Average temperature ranges over which Arrhenius plot of the heat-induced currents was linear and used for the estimation of the maximum *Q*_10_. Horizontal grey area indicates the upper and lower limits of the temperature range for wild-type channels. (**E**) The probabilities obtained from the *t*-tests that were performed in order to determine if there was a significant difference between the *Q*_10_ (white vertical bars) and the lower temperature limit for the maximum *Q*_10_ estimation (grey vertical bars) for the wild-type and the individual mutants. The level of significance *p* = 0.01 is indicated with a dotted horizontal line.

**Figure 5 ijms-20-03990-f005:**
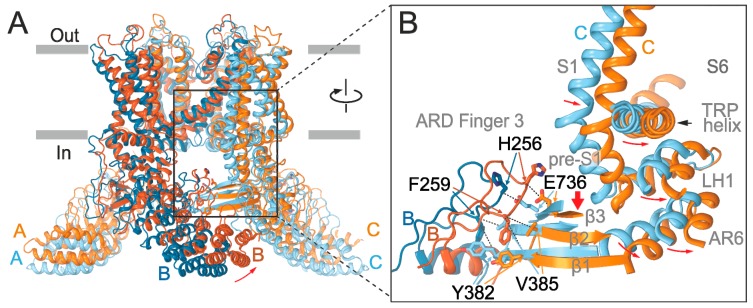
Changes in the β sheet-ARD finger 3 inter-subunit interface during TRPV3 gating. (**A**) Superposition of TRPV3 structures in the closed (PDB ID: 6DVW; subunits A and C are colored cyan, B blue) and open (PDB ID: 6DVZ; subunits A and C are colored orange, B brown) states. The structures are aligned based on their pore domains. The upward movement of the ARD during opening is illustrated with the red arrow. (**B**) Close up view of the β sheet-ARD finger 3 interface between subunits B and C. Sliding down motion of the three-stranded β sheet relative to the ARD finger 3 during TRPV3 opening is illustrated with the thick red arrow. Thin red arrows illustrate movements of the domains that link β sheet to the gate formed by S6. Flexible loops connecting these domains are omitted for clarity. Distances between Cα atoms of cysteine-substituted pairs F259/Y382, F259/V385, and H256C/E736C (black dashed lines) are 8.3 Å, 6.6 Å, and 7.3 Å in the closed state and 9.6 Å, 7.3 Å, and 8.1 Å in the open state, respectively.

**Figure 6 ijms-20-03990-f006:**
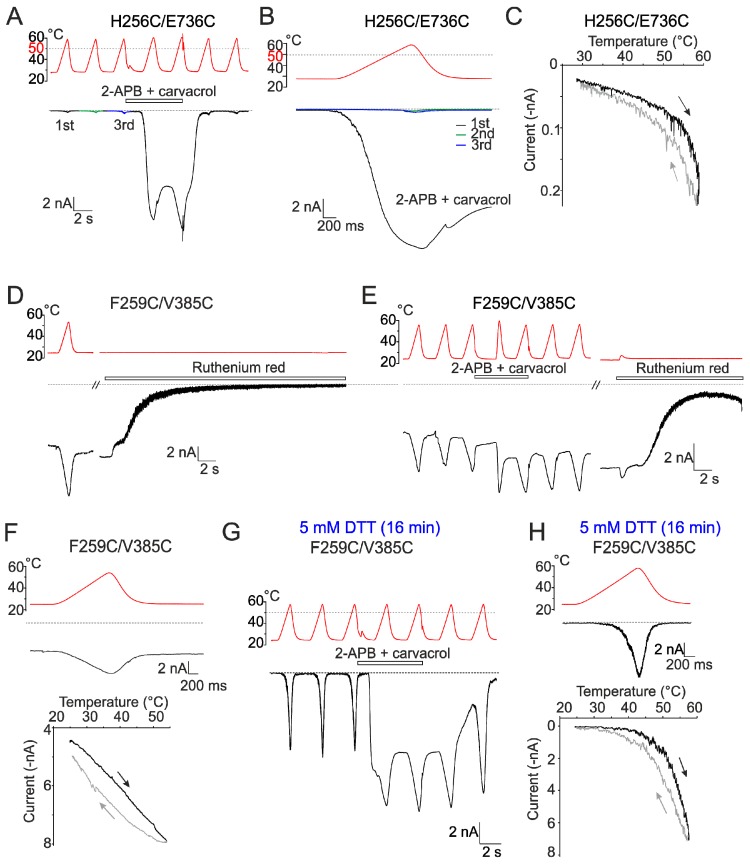
Disulfide locking experiments. (**A**) Representative whole-cell currents evoked by repetitive heat stimuli (shown above the records) recorded from HEK293T cell expressing the H256C/E736C double cysteine mutant channels. Currents were recorded in the absence or presence of mixed agonists (100 µM 2-APB + 100 µM carvacrol), as indicated by the horizontal bar above the records. Holding potential, −70 mV. (**B**) Expanded traces from A. (**C**) Current–temperature relationship of the first heat response from the recording shown in A and B. (**D**,**E**) Representative whole-cell currents recorded at −70 mV, evoked by heat stimuli, recorded from the indicated double cysteine mutants of TRPV3. The channels exhibited large basal currents that were fully blocked by ruthenium red (10 µM). The F259C/V385C double cysteine mutant exhibited strongly sensitized phenotype. (**F**) Expanded trace (top) and current-temperature relationship (bottom) of the first heat response shown in D. Notice a nonlinearity of the deactivation phase. (**G**) Representative F259C/V385C-mediated currents evoked by repetitive heat stimuli recorded from HEK293T cell pretreated with 5 mM dithiothreitol (DTT) for 16 min. (**H**) Expanded trace (top) and current–temperature relationship (bottom) of the first heat response shown in G.

**Figure 7 ijms-20-03990-f007:**
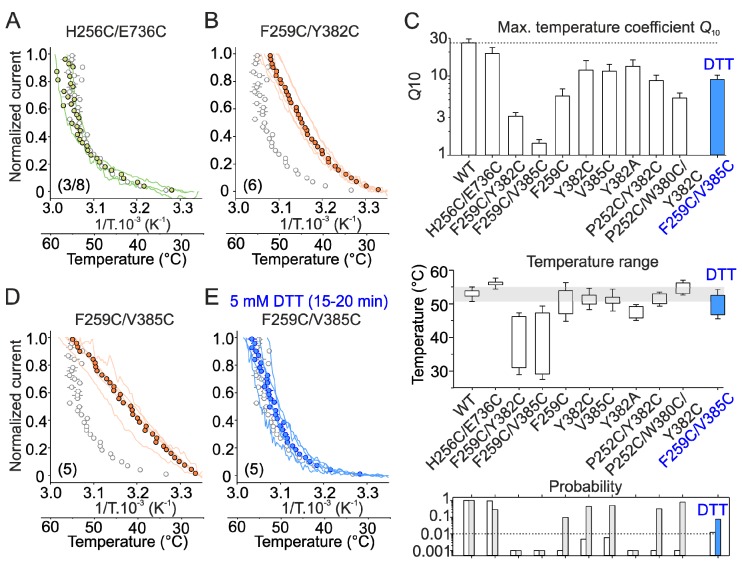
Statistical evaluation of maximum apparent *Q*_10_ and the estimated temperature thresholds for the cysteine mutants. (**A**,**B**) Heat responses from HEK293T cells expressing the indicated double cysteine mutants, normalized to the maximum amplitude with average values overlaid (colored circles with gray error bars indicating S.E.M.). The average current for wild-type TRPV3 is overlaid for comparison as empty circles with grey bars indicating mean ± S.E.M. Notice, that in the H256C/E736C mutant, only 3 out of 8 cells could be included in the statistics. (**C**) Summary of the maximum temperature coefficient *Q*_10_ values for the measured mutants, compared with wild-type TRPV3 (WT). The horizontal dotted line indicates average maximum *Q*_10_ of WT. Middle, average temperature ranges over which Arrhenius plot of the heat-induced currents was linear and used for the estimation of the maximum *Q*_10_. Horizontal grey area indicates the upper and lower limits of the temperature range for WT. Below, the probabilities obtained from the *t*-tests that were performed in order to determine if there was a significant difference between the *Q*_10_ (white vertical bars) and the lower temperature limit for the maximum *Q*_10_ estimation (grey vertical bars) of WT and the individual mutants. The level of significance *p* = 0.01 is indicated with a dotted horizontal line. Notice the narrow temperature interval used to estimate the parameters for the H256C/E736C mutant (**C**, middle panel). (**D**,**E**) Normalized heat responses from the F259C/V385C mutant measured from untreated cell (**D**) and from cells pretreated with dithiothreitol, DTT, for 15–20 min (**E**). The average current for WT is overlaid for comparison as empty circles with grey bars indicating mean ± S.E.M. The number of cells is indicated in parentheses.

**Figure 8 ijms-20-03990-f008:**
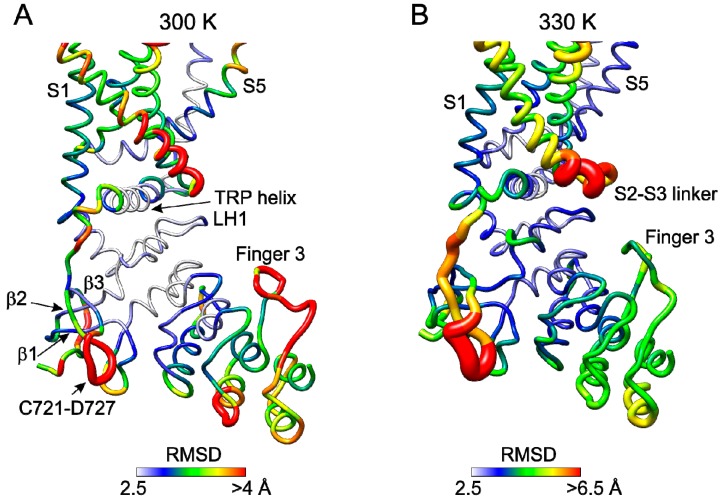
Molecular flexibility of TRPV3 at 300 and 330 K. (**A**,**B**) Results from 50-ns MD simulations of the closed-state TRPV3 structure (PDB ID: 6DVW) at 300 K (~27 °C, A) and 330 K (~57 °C, B). The RMSD values calculated for the backbone atoms are shown for one monomer (chain A). The most flexible residues are colored red (RMSD >4 Å at 300 K and >6.5 Å at 330 K), intermediately flexible yellow-to-green (>3.2 Å and >4.5 Å) and the least flexible blue-to-white (<2.9 Å and <3.5 Å). The worm radius is proportional to RMSD (2–8 Å at 300 K and 2–10 Å at 330 K).

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
