# Peer review of "Cytoplasmic Inter-Subunit Interface Controls Use-Dependence of Thermal Activation of TRPV3 Channel"

_ijms, 2019, doi:10.3390/ijms20163990_

Round 1
Reviewer 1 Report
in the manuscript by Macikova et al. the authors present evidence that interaction of the ARD finger 3 tip with the neighboring subunit β-sheet of TRPV3 channels is responsible for its temperature dependence. Moreover, mutation of a single residue E376 in this region generated a heat-resistant TRPV3 channel. These results were obtained by electrophysiological characterization of TRPV3/V1 chimeric, introduction of single point mutations and molecular modelling. The data are novel, clear cut and presented in an excellent way.
Minor Points of criticism:
(1) I would only recommend to be more carefully in describing TRPV3 activators and inhibitors, because 2-APB, cavacrol and ruthenium red are not specific for TRPV3. This topic needs to be discussed and references should be provided that these substances act not on native channels in untransfected HEK293T cells .
(2) The TRPV3/V1 chimera obviously induce currents in the plasma membrane, but it would further improve the quality of the manuscript, if some evidence for plasma membrane localization would be provided (e.g. Western Blotting of membranes, staining of membranes with fluorescent-coupled specific TRPV3 antibodies in confocal images or surface biotinylation).
(3) Many experiments and important conclusions unfortunately do not appear in the abstract. Authors should try to mention more specific results even if space is limited.
Reviewer 2 Report
The work by Macikova et al. is a very elegant way to discover a heat responsive structure in the vanilloid channel TRPV3. The authors investigated the current high resolution structures of TRPV channels and compared a finger 3 structure located in the N-terminus of the channel, between two ankyrin repeats. Interestingly, this short non-conserved sequence is within a highly conserved region. The authors used repetitive heat ramps and recorded the activity of TRPV3 channels and chimeras as well as mutations of TRPV1 non-conserved finger 3 segment. The substitution of this finger 3 structure has large effects on TRPV3 channel temperature sensitivity and shifts the activation to lower temperatures. A detailed screen for essential amino-acids within the finger 3 segments clearly identified a proline 252 as key residue. A huge amount of patch clamp experiments were performed in this manuscript and the results are of very high quality. Also the accurate heat protocol as used in this work is important. The authors combined recent high resolution structures with a cysteine disulfide approach. The TRPV3 structure revealed that finger 3 contributes to the intersubunit interface between the ankyrin repeat domain and the three-stranded β-sheet composed of a β -hairpin from the N-terminal MPD linker and a C-terminal β -strand, both from the adjacent subunit. In line with the functional importance of this structure cysteine substitutions in both domains (H256C/E736C) largely shifted temperature activation to degrees above 60°C. I’d like to note that the authors used perfect controls of TRPV3 activation in the absence of temperature: 2-APB and carvacrol. Other combined cysteine mutations locked the channel in an open configuration. Again here the authors used ideal controls of 5mM DTT to interfere with cysteine crosslinking. Hence, the authors convincingly conclude that interaction of the finger 3 structure with the neighboring subunit β-sheet can alter the open channel state of TRPV3 channel and affect the temperature dependence. These results also determine important structural and mechanistic insights into temperature dependent gating of the TRPV3 channel.
This manuscript presents very compelling evidence, supported my solid patch clamp data to determine the role of a N-terminal finger domain in TRPV3 for temperature dependent channel gating. Few minor weaknesses are noted, but they can be fixed fairly easily.
Minor point: The authors mention that cells expressing F259C/V385C were so deleteriously leaky and present a constitutively active mutant. Please provide IV curves of important TRPV3 double cysteine mutants in comparison to wild-type TRPV3.
The MD simulations are just 10ns long, if the computer network allows for longer traces I would recommend to run the simulations up to 100ns.
MD simulations should be described more detailed in the methods section.
